

# The weathering process of carbonatite: weathering time

Jin Chen[1], Fangbing Li[1], Xiangwei Zhao[1], Yang Wang[1], Limin Zhang[2], Ling Feng[1], Xiong Liu[1], Lingbin Yan[1] and Lifei Yu[1]

[1] Key Laboratory of Plant Resource Conservation and Germplasm Innovation in Mountainous Region, Collaborative Innovation Center for Mountain Ecology & Agro-Bioengineering (CICMEAB), College of Life Sciences/Institute of Agro-bioengineering, Guizhou University, Guiyang, Guizhou, China
[2] Institute of Guizhou Mountain Resources, Guizhou Academy of Sciences, Guiyang, Guizhou, China

## ABSTRACT

Soil formation by rock weathering is driven by a combination of parent material, climate, organisms, topography, and time. Among these soil-forming factors, time plays a pivotal role in the weathering of carbonatite but it is a challenging factor to study quantitatively. A method for determining the weathering duration of carbonatite based on its weathering characteristics over a century-scale time period has not been clearly established. In this study, we selected abandoned carbonatite tombstones commonly found in the karst region of southwest China for investigation, using the date when the tombstones were erected as the onset of weathering. Chemical weathering indices were used to evaluate the weathering degree of different oxide contents produced by the carbonatite weathering process. In order to explore the weathering characteristics over time, the relationship between weathering duration and weathering degree was established. The results showed the following: (1) magnesium (Mg), aluminum (Al), silicon (Si), iron (Fe), titanium (Ti) are gradually enriched in the carbonatite regolith, and calcium (Ca) is gradually reduced. (2) The chemical indices of alteration (CIA), leaching coefficient (Lc), alumina-to-calcium ratio (AC) and mobiles index ($I_{mob}$) can be successfully used for evaluation of the weathering degree of the carbonatite in different weathering time periods. (3) During the weathering of carbonatite, the weathering rate is a logarithmic function of time. Our research shows that over a period of more than 100 years of weathering, the carbonatite weathering process is characterized by fast weathering rates and low weathering degree in the early stages, but slow weathering rates in the later stages.

Corresponding author
Lifei Yu, gdyulifei@163.com

## INTRODUCTION

Carbonatite is an important component of rock in karst areas worldwide (*Goldscheider et al., 2020*). It is composed mostly of calcite with small admixtures of other minerals. It therefore undergoes specific weathering processes consisting mainly of Ca dissolving and leaching (*Noiriel et al., 2009*; *Emmanuel & Levenson, 2014*). In the natural environment, the carbonate minerals in carbonatite react with water and dissolved carbon dioxide ($CO_2$) to produce Ca (*Liu et al., 2020*), Mg and bicarbonate ions, which is ultimately the
reason for the formation of karst (*Shumka & Ciftci, 2018*). When dissolution occurs, the micron-sized particles on the surface of carbonatite begin to fall off, thereby accelerating the weathering of the rock surface (*Emmanuel & Levenson, 2014*; *Levenson & Emmanuel, 2016*). The rock area where the dissolution takes place forms a grike, and the remaining part forms a clint (*Jones, 1965*). Subsequently, leaching, hydration, carbonation and oxidation processes occur on the surface of the carbonatite (*Zha et al., 2020*). In urban areas, gypsum formation may also occur due to the interaction of atmospheric sulfur dioxide ($SO_2$) and calcite from the stone (*Saheb et al., 2016*). Carbonatite contains a low proportion of weathering residues (about 5%) (*Sun et al., 2002*), has a slow soil formation rate (about $2.5-5.0\ \mu m/a$) and lacks nutrient elements. The soil is generally eroded by running water faster than it can be formed. Rocky desertification often occurs because both soil and water conservation are weak (*Yuan, 1997*; *Wang et al., 1999*; *Chen et al., 2014*). In karst areas, therefore, weathering has important ecological implications for ecosystem construction, maintenance of species diversity (*Rahbek et al., 2019*), and carbon sequestration (*Li et al., 2019*). In addition, the degree of rock weathering is related to the engineering properties of the rock material (*Lumb, 1962*). Hence, a simple and rapid method for determining the degree of rock weathering is required.

Three methods can be used to evaluate the relationship between rock weathering rate and weathering time (*Colman, 1981*): (1) laboratory research, involving precise control of weathering time and rock weathering process; (2) study of man-made features such as tombstones and buildings, using historical records to control time; and (3) study of naturally occurring sediments or materials, using independent age estimates. The large uncertainty in the weathering rate of carbonatite in field environments is one of the difficulties in assessing its dissolution. Many field experiments last only a few years, and cosmogenic isotope methods capable of providing long-term rate estimates have so far been of limited value. Generally, the weathering of building materials and rocks in nature is quantified as the loss of certain mechanical properties measured by a series of laboratory destructive tests (such as uniaxial compression tests and point load tests) (*Ercoli et al., 2013*; *Levenson & Emmanuel, 2016*). However, in some cases, these methods are not suitable. An example would be in the areas of stone construction and cultural relic protection, where the tests could cause damage. An additional method allows the degree of weathering to be measured by a chemical weathering index of the main oxides contained in the rock (*Okewale, 2020*). Although the weathered part of carbonatite usually dissolves, the rock also undergoes a certain degree of oxidation due to the presence of metals such as Fe, Al, *etc.* (*Zha et al., 2020*). Using this type of index to measure the weathering degree of carbonatite may, therefore, be a new approach worth trying.

Chemical weathering indices based on rock oxides have been used to examine the weathering patterns of various rock types. Studies have established patterns and models for the degree of weathering *versus* depth (*Okewale & Coop, 2018*; *Okewale, 2020*), and likewise for weathering in relation to time (*Colman, 1981*). These models are ubiquitous and provide new insights for understanding the mechanisms, which can then help to build a theoretical framework for the processes involved in rock weathering into soil. Recent studies have explored a variety of weathering patterns including those for magmatic rocks,

siliciclastic rocks, granites, basalts, gneisses, and others. They have used various oxide-based chemical indices (*Vogt, 1927*; *Reiche, 1943*; *Ruxton, 1968*; *Parker, 1970*; *Harnois & Moore, 1988*; *Jayawardena & Izawa, 1994*; *Irfan, 1996*) and have proven to be a powerful tool for studying the degree of rock weathering. Nevertheless, evaluating the variation in rocks with different properties over time has always been challenging. On the one hand, among the five soil-forming factors, time is extremely important yet surprisingly difficult to quantitatively evaluate. On the other hand, the chemical properties of different parts of the same rock may differ, rendering it problematic to use a given index for evaluation uniformly. Moreover, insufficient research effort has been directed at deriving a model that accurately evaluates the relationship between carbonatite's weathering time and degree at the century scale. By establishing a relationship for the carbonatite of karst areas, this study will help advance such research.

Southwest China's karst area is one of the most widely distributed and concentrated areas of karst terrain in the world, featuring large areas of exposed carbonatite. The rock is, therefore, widely used for local building and is the main material used for tombstones. According to local customs, the year of a tombstone's erection is engraved on it, thus providing a relatively accurate start time for the weathering process. Considering the possible damage from sampling, we chose to conduct investigation on abandoned tombstones (tomb relocation events have occurred). We took the date engraved on the abandoned tombstone as the weathering start time, and calculated the weathering duration of the carbonatite (calculated as 2021 minus the engraved year on the tombstone). The study uses these different known exposure durations to help us better understand the processes and mechanisms of carbonatite weathering. In particular, we addressed the following questions: (1) To what extent is the oxide-based chemical weathering index suitable for assessing the weathering of carbonatite, where dissolution is the primary process? (2) How does the degree of weathering change with the increase of weathering time of carbonatite?

In order to answer these questions, we selected carbonatite tombstones concentrated in a small area in China showing little disturbance. We applied a chemical index to evaluate their weathering degree, and then examined the relationship between the carbonatite's weathering time and its degree of weathering. We also assessed the applicability of the AC index, which was developed specifically for carbonatite weathering. Our research will provide some new insights for the restoration of degraded ecosystems in karst areas.

## MATERIALS AND METHODS

### Site description and sampling strategy

The sampling site was located in the Central Guizhou karst zone (forestland in the southern part of the Xibei community, Huaxi District, Guiyang City, 26° 26′46″−−26°26′56″, 106°38′06″–106°39′18″), in the concentrated karst area of Southwest China. Here, high levels of water and heat coincide, with an average annual air temperature of 15.7 °C, an average annual rainfall of about 1,100 mm, and annual cumulative sunshine duration of about 1000—1100 h; https://www.huaxi.gov.cn/lxhx/zrdl/). Due to both geography and local customs, many carbonatite tombstones are distributed throughout this area. From our

preliminary investigation, we found 10 suitable abandoned carbonatite tombstones. The erection period of these abandoned tombstones was from 1860 to 1999, so the weathering time was from 22 to 161 years 114 (specifically 22, 23, 37, 39, 58, 72, 88, 104, 133 and 161 years).

Carbonatite contains magnesium carbonate ($MgCO_3$) and calcium carbonate ($CaCO_3$), so it will foam violently when 5% cold dilute hydrochloric acid is applied. To ensure that the tombstones we chose were true carbonatite, we relied on the results of this cold acid test and on subsequent X-ray diffraction (XRD) tests combined with other weathering indicators. If they were definitely carbonatite, then the date of erection and other basic information (such as slope, aspect, latitude and longitude) was recorded. Each sampling point was also numbered. Data were collected as previously described in *Chen et al. (2022)*, specifically the sampling strategy.

The abandoned tombstones are located deep in the forest, so they are less likely to be exposed to air pollution. In addition, moss and/or lichens (Table S1) were removed as previously described in *Chen et al. (2022)*. During sampling, it was difficult to collect the regolith on the E surface of the abandoned carbonatite tombstone with hand tools. We therefore used an angle grinder to do the sampling. To obtain a true weathered crust sample, we controlled the angle grinder to ensure full contact with the rock until an entire available weathered surface was removed. To prevent cross-contamination, we replaced the grinding disc for each new tombstone. In addition, as a control for the regolith, we used a geological hammer to cut it back and obtain a sample of the un-weathered abandoned carbonatite tombstone. Detailed sampling methods are shown in *Chen et al. (2022)*.

## Measurement methods

We used powder XRD to determine the phase of the parent rock. The scanning speed was 2°/min, the scanning angle 5−90°, and the test target was cuprum (Cu). The oxide content in the weathered crust samples and parent rock samples was determined by X-ray fluorescence (XRF) spectrometry, using a Zetium XRF spectrometer from PANalytical, with an error range of ± 0.0001. The sample powder was placed in the instrument, using the press method, in oxide mode, with sequential scanning, 4 kW power, a maximum voltage of 60 kV and a maximum current of 170 mA. The element detection range was Be (4) - U (92) and the detection limit was $10^{-6} \sim 100\%$. Prior to this, in order to prevent the potential organic matter content in the sample from affecting the test results, we placed the sample in a porcelain crucible and heated it to 950 °C.

## Selection of a chemical weathering index

The essence of rock weathering is elemental migration and accumulation, so the degree of migration and accumulation can be used as a proxy for weathering. The one commonly used is the weathering coefficient, which is the ratio of two or more oxides. Because the value of this weathering coefficient entirely characterizes the rock, it is also called the absolute weathering rate (also sometimes chemical index of alteration, chemical index of weathering, *etc*). When comparing different parent rocks with a wide range of properties, there will be a large deviation in the absolute weathering rate when characterizing the

**Table 1** Details of the chemical weathering indices evaluated for limestone in this study.

| Index | Formula | References |
|---|---|---|
| Alumina-to-calcium ratio (AC) | $Al_2O_3/CaO$ | This study |
| Chemical index of alteration (CIA) | $100\times[Al_2O_3/(Al_2O_3+CaO+Na_2O+K_2O)]$ | *Nesbitt & Young (1982)* |
| Leaching coefficient (Lc) | $SiO_2/(K_2O+Na_2O+CaO+MgO)$ | *Li et al. (1995)* |
| Mobiles index ($I_{mob}$) | $[(K_2O+Na_2O+CaO)_{fresh}-(K_2O+Na_2O+CaO)_{weathered}]/(K_2O+Na_2O+CaO)_{fresh}$ | *Irfan (1996)* |
| Weathering strength (Ws) | $Ws= 100\times[1-(E_1\times ave(t)+E_2)/2]$ $E_1$ is the degree of equilibrium of the three absolute weathering rates ($SiO_2/Al_2O_3$, $Al_2O_3/Fe_2O_3$, $R_2O_3/SiO_2$); $E_2$ is the equilibrium degree of the variation coefficient of seven oxides ($SiO_2$, $Al_2O_3$, $Fe_2O_3$, CaO, MgO, $K_2O$, $Na_2O$); ave(t) is the average value of the leaching coefficient $t$ of five oxides ($SiO_2$, CaO, MgO, $K_2O$, $Na_2O$), in this formula, $t = 100\times(t_1-t_2)/t_1$; $t_1$ is the weight percentage of the oxide in parent rock; The formula for calculating $t_2$ is: $100\times t_3\times(Al_2O_3)_{fresh}/(Al_2O_3)_{weathered}$; $t_3$ is the weight percentage of the oxide in regolith; The formula for calculating equilibrium degree, E, is: $E=e^{H(s)}/S$, where, $H(s) =-\sum_{i=1}^{s}p_i\times\ln(p_i)$; in this formula, $p_i= n_i/N$, for which $N$ is the sum of a set of values, $n_i$ is the value of the $i$-th number, and $\ln(p_i)$ is the natural logarithm of $p_i$. | *Huang et al. (1996)* |

weathering degree. A weathering rate relative to the parent rock is, therefore, required to accurately characterize the weathering degree of the rock, *i.e.*, the relative weathering rate (such as the $I_{mob}$). Both absolute and relative weathering rates are calculated according to molecular ratios (Table 1). A comprehensive weathering indicator, such as a scoring system and weathering strength (Ws) index are also used to fully evaluate the weathering degree of rocks.

The karst geological system is relatively complex, and a single weathering index may not fully convey its complexity (*Okewale & Coop, 2018*). The SA index measures decreasing content of silica, which is an admixture in calcareous rocks (*Gupta & Rao, 2001*). This index can give some information about processes occurring in calcareous rocks, but not the main process of $CaCO_3$ dissolution. The CIA (*Nesbitt & Young, 1982*), chemical indices of weathering (CIW) (*Harnois & Moore, 1988*) and plagioclase alteration (PIA) (*Fedo, Wayne Nesbitt & Young, 1995*) are used for the conversion of feldspars to clays (*Okewale, 2020*). Those indices could be used for carbonatite weathering rates, similar to Vogt's residual index (VRI) (*Vogt, 1927*), Alumina to calcium-sodium ratio (ACN) (*Harnois & Moore, 1988*) and Lc (*Li et al., 1995*). Inspired by this, we decided to try to describe the dissolution process of carbonatite using the simplified alumina-to-calcium ratio (AC, $Al_2O_3/CaO$). Finally, the mobiles index $I_{mob}$ (*Irfan, 1996*) characterizes the elemental migration of the weathered layer relative to the parent rock layer during the weathering of carbonatite, which gives a relative measure of the removal of mobile cations by rock weathering and can also be used for the dissolution process of carbonatite. In addition, we also characterized the weathering process of carbonatite with the weathering intensity, which is a comprehensive weathering index (*Huang et al., 1996*). The detailed calculation methods of these weathering indicators are shown in Table 1.

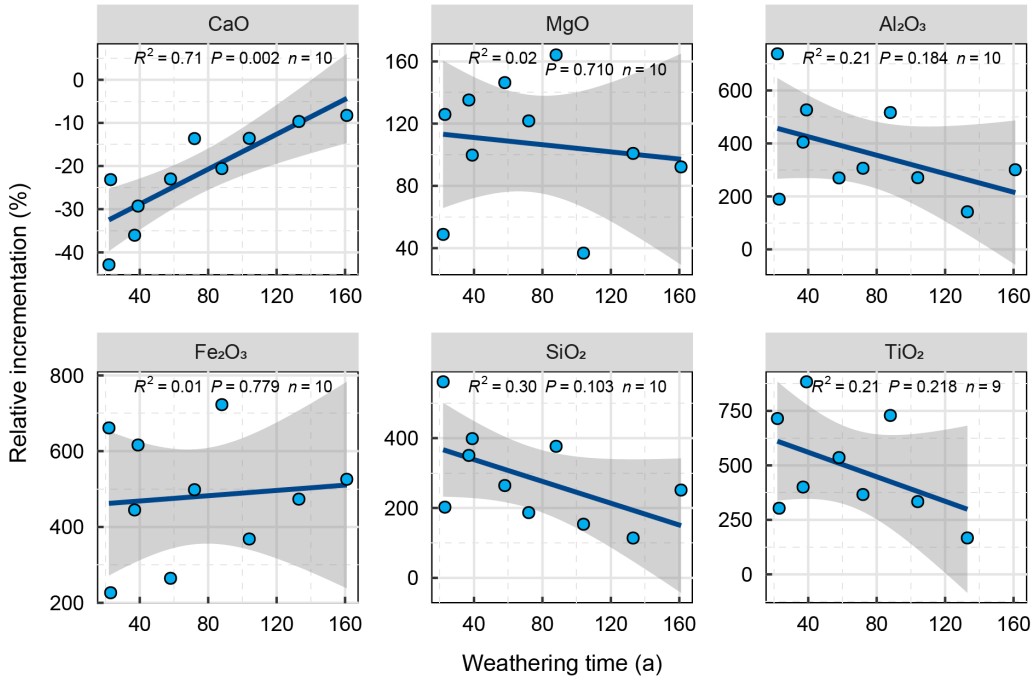

**Figure 1** **Relative variation of elements in carbonatite weathering crust with weathering time.** The shaded areas around the fitted regression line represent the 95% confidence intervals, $R^2$ represents model fit degree and *p*-value represents model significance degree. a; annual.

## Data analysis

The relative incrementation of oxides of the main metal elements in the regolith $= 100\% \times (M_r w - M_r p)/M_r p$, where $M_r w$ is the molecular ratio of the metal element in the weathering crust, while $M_r p$ is the molecular ratio of the metal element in the parent rock. In order to analyze the variation characteristics of each weathering index with weathering time, the 'lm' function of the built-in statistics package in R was used to test the significance of the linear model, and the *ggplot2* package was used for plotting.

## RESULTS

### The differences in oxides between regolith and parent rock

Based on reports by *Chen et al. (2022)*, the samples we collected were all carbonatite. Because the properties of the different carbonatite parent rocks vary, the change of element content in the weathering layer relative to the parent rock layer is a better reflection of the change of the main elements of carbonatite over different weathering durations. The XRF results show that, relative to the parent rock, the elements Mg, Al, Si, Fe, and Ti in the weathering crust are gradually enriched (relative increment is greater than 0) with weathering duration. However, relative to the parent rock layer there is no significant change in the increase. Ca gradually decreases (relative increment is less than 0), and the decrease relative to the parent rock increases significantly with the increase of weathering time (Fig. 1).

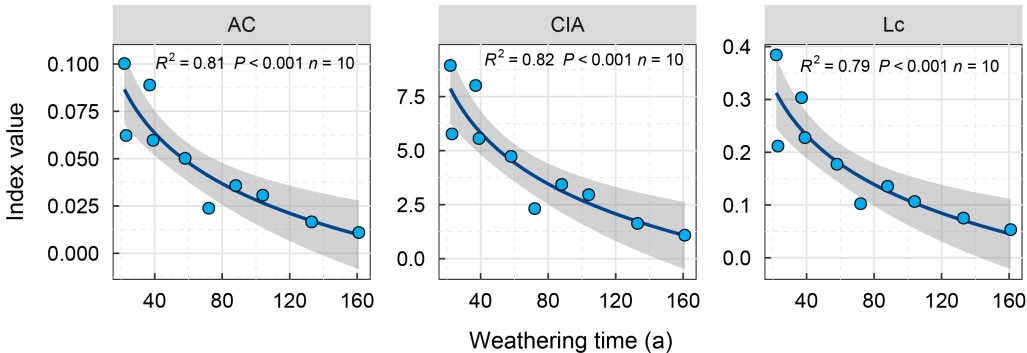

**Figure 2** **Regression models for absolute weathering rate as a function of weathering time.** The shaded areas around the fitted regression line represent the 95% confidence intervals. a; annual.

## The association between absolute weathering rate and weathering time

The absolute weathering rate is the weathering value that characterizes the regolith itself, *i.e.,* the degree of migration and accumulation of elements in the regolith. A suitable index should be able to give clear trends and meaningful statistics, as well as being sensitive to the control process. We used the AC index (the proportion of alumina and calcium oxide) to characterize the dissolution of soluble material in the weathering of carbonatite. The greater the AC, the stronger the weathering rate. The AC index decreased significantly with the weathering duration (Fig. 2), also indicating that the weathering rate was fast in the early stages and slower in the later stage. The CIA index and Lc index were used to characterize the process of feldspar transformation into clay during rock weathering. In a similar way to the AC index, both CIA and Lc decreased significantly with weathering duration. However, considering the fact that in carbonatite Ca is much more abundant than other basic cations, it is not surprising that all these indices had a very similar pattern. In conclusion, the AC index shows good applicability for evaluating the weathering degree of carbonatite in different weathering time periods.

## The association between relative weathering rate and weathering time

The relative weathering rate is obtained from measurements of element migration and accumulation relative to the parent rock. This gives a relative measure of the removal of mobile cations (*Irfan, 1996*), therefore, we use a mobiles index to characterize the weathering rate of carbonatite relative to its parent rock during weathering (Fig. 3). With the increase of weathering duration, the mobile cations produced by the weathering of carbonatite relative to the parent rock gradually decreased (Fig. 3). This also indicated that the weathering was more intense in the early stages and gradually weakened in the later stages. Based on the relationship between mobiles index and weathering time, we get the regression equation between them. Based on the regression equation, when the weathering time of carbonatite reaches about 373 years, the migration index tends to 0, which may mean that the weathering of carbonatite regolith is suspended. The absolute weathering rate and relative differentiation rate both describe aspects of weathering intensity, but too many indicators may obscure each other, and the generalized comprehensive index better

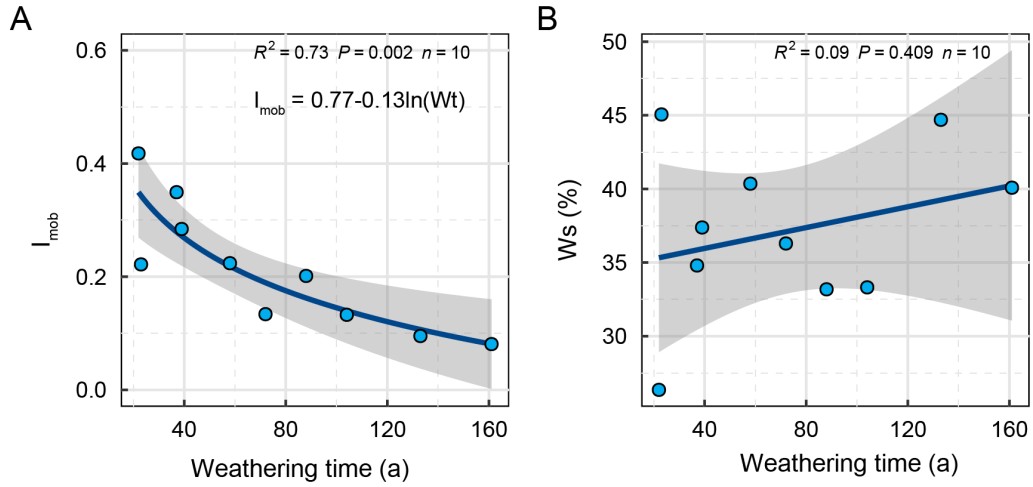

**Figure 3 Relationships of relative weathering rate (A) and comprehensive weathering index (B) to weathering time.** The shaded areas around the fitted regression line represent the 95% confidence intervals. a; annual.

reflects the weathering intensity relative to the parent rock. The integrated weathering intensity increased with weathering duration (Fig. 3), but did not reach a significant level. According to the absolute weathering index (Fig. 2) and the relative weathering index (Fig. 3), it is clear that the weathering rate of carbonatite is fast in the early stages and slower in the later stages, while the degree of weathering is small in the early stages and larger in the later stages.

## DISCUSSION

### The applicability of a chemical weathering index to the weathering process of carbonatite

Carbonatite weathering is a process that enriches the elements Al, Fe Mg Si and Ti while reducing Ca (Fig. 1). The weathering of carbonatite is often thought to be controlled by chemical dissolution and can occur relatively uniformly or selectively. The main way to evaluate the degree of weathering of carbonatite is, therefore, to couple the degree of weathering with either the mechanical mechanisms, microscopic observation of the weathered surface (*Zha et al., 2020*), or determinations of the rock bulk density or specific volume (*Okewale, 2020*). As the main process taking place during carbonatite weathering is dissolution, weathering rates are usually measured as a difference in rock mass over time (*e.g., Thorn et al. (2002); Wasak-Sęk, Jelonkiewicz & Drewnik (2021)*) or rock surface recession in comparison to a reference surface, *e.g.,* the lower part of the slab (*Feddema & Meierding, 1987; Inkpen & Jackson, 2000*). However, carbonatite also undergoes oxidation where it contains metallic elements such as Fe and Al (*Zha et al., 2020*). After dissolution, the remaining grikes and clints provide places for the oxides to accumulate. In a similar way to the CIA and Lc indices for conversion of feldspar into clay, the AC index we used has good applicability to the characterization of the weathering degree of carbonatite after

different weathering durations (Fig. 2). In addition, the relative weathering rate ($I_{mob}$) has good applicability to characterize the weathering rate of carbonatite. Finally, some studies have shown that weathering strength increases with the increase of weathering time (*Chen et al., 2022*). In this study, there is no significant change trend between Ws and weathering time, which may be caused by the sample size limitation.

### Variation trend of the weathering rate of carbonatite with weathering time

In Changle, Fujian Province, China, under long-term erosion by water, the weathering rate of a granite pit was about 10.8 mm/100 yr over a period of more than 40 years (*Wang, Huang & Feng, 2020*). The vertical plane of the carbonatite monument in Louisville, USA, is dissolving at a rate of about 0.5 mm/100 yr (*Meierding, 1993*). However, these results came from experiments which all measured the rock surface changes after a specific weathering duration, and did not monitor the entire weathering process. Studies have shown that the relationship between the decay loss of marble tombstones and the weathering time can be described as a power function relationship (*Inkpen, Mooers & Carlson, 2017*) or a linear relationship (*Inkpen & Jackson, 2000*). Whatever the type of relationship, the studies showed that the degree of weathering increases with greater time. There are various relationships between the rock weathering rate and the weathering duration. When the rock undergoes congruent dissolution, the weathering rate is a linear function of the weathering time. However, the weathering of mineralogically heterogeneous rocks involves complex physical and chemical processes that can usually only be represented by logarithmic time functions (*Colman, 1981*).

When the minerals that make up the rock are heterogeneous, the stable residues produced during the weathering process may change the path of running water, reducing the loss of elements in some areas, protecting un-weathered areas to a certain extent, and slowing down the rate of weathering generally (*Colman, 1981*). Typically, long duration is needed (*e.g.*, 0.5 Myr for American volcanic rocks) to allow the residues to equilibrate (*Colman & Pierce, 1981*). For a considerable period of time, therefore, the rate of debris formation far exceeds the rate at which it inhibits weathering, so it is not surprising that the rate of weathering is a logarithmic function of time. In this study, the weathering process of carbonatite was approximately a logarithmic function of time. Congruent dissolution occurs only with high purity carbonatite and the purity of the discarded tombstones rock is not high enough. Therefore, when the carbonatite is dissolved and oxidized, some insoluble residues are still produced. Grikes and clints created by the dissolution then provide an accumulation place for the residues. Carbonatite undergoes a weathering process similar to that of mineralogically heterogeneous rocks, resulting in a weathering rate that is approximately a logarithmic function of time.

## CONCLUSION

This study explores the weathering characteristics of carbonatite on centennial timescales. We have three main conclusions. Firstly, although the weathering process of carbonatite is mainly dissolution, it also undergoes other processes. Secondly, the chemical weathering

index based on regolith oxides is applicable to the measurement of carbonatite with different weathering durations, especially the AC index developed for carbonatite. Finally, the weathering rate gradually decreases with increased weathering time, while the degree of weathering increases. These conclusions provide a new perspective for understanding the weathering process of carbonatite.

## ACKNOWLEDGEMENTS

We would like to thank the editors and reviewers for their selfless help. In addition, Jin Chen wants to thank, in particular, the patience, care and support from Yan Yao over the past 9 years.

### Funding

This work was supported by the 13th Five-year National Key Research and Development Plan (grant number 2016YFC0502604), the Construction Program of Biology First-class Discipline in Guizhou (grant number GNYL[2017]009), and the Postgraduate Education Innovation Program in Guizhou Province (grant number YJSKYJJ[2021]079). The funders had no role in study design, data collection and analysis, decision to publish, or preparation of the manuscript.

### Grant Disclosures

The following grant information was disclosed by the authors:
13th Five-year National Key Research and Development Plan: 2016YFC0502604.
Construction Program of Biology First-class Discipline in Guizhou:  GNYL[2017]009.
Postgraduate Education Innovation Program in Guizhou Province:  YJSKYJJ[2021]079.

### Competing Interests

The authors declare there are no competing interests.

### Author Contributions

- Jin Chen conceived and designed the experiments, performed the experiments, analyzed the data, prepared figures and/or tables, authored or reviewed drafts of the article, and approved the final draft.
- Fangbing Li conceived and designed the experiments, performed the experiments, analyzed the data, prepared figures and/or tables, authored or reviewed drafts of the article, and approved the final draft.
- Xiangwei Zhao conceived and designed the experiments, performed the experiments, analyzed the data, prepared figures and/or tables, authored or reviewed drafts of the article, and approved the final draft.
- Yang Wang conceived and designed the experiments, performed the experiments, analyzed the data, prepared figures and/or tables, authored or reviewed drafts of the article, and approved the final draft.

- Limin Zhang conceived and designed the experiments, performed the experiments, analyzed the data, prepared figures and/or tables, authored or reviewed drafts of the article, and approved the final draft.
- Ling Feng conceived and designed the experiments, performed the experiments, analyzed the data, prepared figures and/or tables, authored or reviewed drafts of the article, and approved the final draft.
- Xiong Liu conceived and designed the experiments, performed the experiments, analyzed the data, prepared figures and/or tables, authored or reviewed drafts of the article, and approved the final draft.
- Lingbin Yan conceived and designed the experiments, performed the experiments, analyzed the data, prepared figures and/or tables, authored or reviewed drafts of the article, and approved the final draft.
- Lifei Yu conceived and designed the experiments, performed the experiments, analyzed the data, prepared figures and/or tables, authored or reviewed drafts of the article, and approved the final draft.

## Data Availability

The raw measurements are available in the Supplementary Files.

## Supplemental Information

Supplemental information for this article can be found online at http://dx.doi.org/10.7717/peerj.15793#supplemental-information.

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
