# Peer review of "The weathering process of carbonatite: weathering time"

_PeerJ, doi:10.7717/peerj.15793_

## Round 0.1 · original submission · Minor Revisions

The submitted manuscript devoted to the application of the chemical index of weathering to evaluate the process of limestone weathering corresponds to the direction of the journal and contains new developments on the subject. Determination and substantiation of chemical weathering indices complement the understanding of the mechanism and rate of rock weathering at its various stages. The research used progressive methods, which in the future can be applied in other studies of a similar topic. I would also like to note the good structure of the manuscript, systematicity and high quality of information presentation. The obtained results have a high applied value for the study of territories with limestone karst. The manuscript is written in high-quality English and can be published in PeerJ. To improve its quality, I suggest responding to the reviewers' wishes and, if possible, taking their opinions into account.

Reviewer 1 ·

Basic reporting

The article titled Applicability of a chemical weathering index in assessing the weathering process of limestone offers an explanation of how quickly limestone weathers. An interesting aspect of the work is the fact that the authors used weathering indices for this purpose. This manuscript is interesting and may interest a wide range of scientists, especially focusing on weathering and carbonate rocks. Nevertheless, I have some comments that could add value to the article.
1. Methods: please add how many samples were analysed. The formulas of individual weathering indices are commonly known, but it is definitely more transparent when the authors include the formulas of these indices in the methods, especially the AC or mobiles index (Imob).
2. Methods: The authors often compare the results of weathering indices in the manuscript relative to the parent rock (e.g. lines 180, 183), but it seems to me that fresh limestone samples from the excavation site and which was used as limestone tombstones were examined. Please explain.
3. Methods: In line 124 the Authors write that "...the tombstones were covered with moss and/or lichens, which was carefully removed...", but they do not refer to this information in the rest of the text. It is not known what percentage of the examined tombstone is covered with moss and/or lichens and what kind they are. What impact can they have on the weathering process and its speed? This requires special attention and supplementation.
4. Results: 4. Results: Lines: 188-192. The authors write "The greater the AC, the stronger the weathering." And that is logical. In a further sentence, they indicate that with the duration of weathering, the value of the AC index decreases. Please explain.
5. Results: The authors repeatedly emphasize that "the weathering rate was fast in the early stages and slower in the later stage", but they do not mention any details after how many years the weathering rate spade process. This requires supplementation, as this statement is used many times in the further part of the work. A figure showing changes in the weathering rate over time could be added, it would summarize the results of the research.
6. Discussion: there is practically no reference to the results of the research presented in the article. Only the authors refer to "the weathering rate was fast in the early stages and slower in the later stage", but also in this case the statements are enigmatic and not very specific. Please complete
7. Add an supplement with output data

In my opinion, the manuscript should be published in PeerJ. However, major changes are required. I recommend that this work is accepted for publication in PeerJ but after major revisions are made.

Experimental design

Yes, it is an experimental design

Validity of the findings

The article may enrich our knowledge about limestone weathering

Additional comments

A figure showing how the weathering proceeds and a data supplement should be added

·

Basic reporting

The title needs modifications as it is not comprehensive and reflects the findings.
line 37: Replace limestone with "it"
Line 37: “It therefore undergoes" Please rewrite this sentence clearly.
line 38, 40, and others, the use of abbreviation and full form are random throughout the manuscript. I will suggest using full form along with an abbreviation inside the parenthesis () at its first use and after that abbreviation could be used throughout the manuscript.
Line 39: Older year should be first.
Line 50-52: sentence formation is weak
Line 59: add "and" before (3)
Line 59 to 64: The sentence-making is poor. Please rewrite these sentences.
line 80: degree of rocks weathering
In the introduction section, the authors describe their adopted methodology, However, it will be better to provide aims and objectives and some short highlights of your finding at the end of the introduction.
Line 122: remove "therefore"
Line 137-138: rewrite the sentence
Discussion is shortly described, Please provide a detailed discussion along with insight mechanisms and a comparison of results with previous literature.

Experimental design

The experiment is conducted carefully and provided sufficient experimental description.

Validity of the findings

In the introduction section, the authors describe their adopted methodology, However, it will be better to provide aims and objectives and some short highlights of your finding at the end of the introduction.
Discussion is shortly described, Please provide a detailed discussion along with insight mechanisms and a comparison of results with previous literature.

---

## Round 0.2 · accepted · Accept

The scientific work devoted to the study of the processes of chemical weathering of rocks is a qualitative and complete study. The work contains important and substantiated data, more broadly reveals the essence of the process of chemical weathering using the example of carbonatite.